# Variants in *MHY7* Gene Cause Arrhythmogenic Cardiomyopathy

**DOI:** 10.3390/genes12060793

**Published:** 2021-05-22

**Authors:** Valentina Ferradini, Luca Parca, Annamaria Martino, Chiara Lanzillo, Elisa Silvetti, Leonardo Calò, Stefano Caselli, Giuseppe Novelli, Manuela Helmer-Citterich, Federica Carla Sangiuolo, Ruggiero Mango

**Affiliations:** 1Department of Biomedicine and Prevention, University of Rome “Tor Vergata”, 00133 Rome, Italy; ferradini@med.uniroma2.it (V.F.); novelli@med.uniroma2.it (G.N.); 2Department of Biology, University of Rome “Tor Vergata”, 00133 Rome, Italy; luca.parca@gmail.com (L.P.); citterich@uniroma2.it (M.H.-C.); 3Division of Cardiology, Policlinico Casilino, 00169 Rome, Italy; martinoannamaria@yahoo.it (A.M.); chiaralanzillo@hotmail.com (C.L.); elisa.silvetti@tiscali.it (E.S.); leonardocalo.doc@gmail.com (L.C.); 4Cardiovascular Center Zurich, 8091 Zurich, Switzerland; stefanocasellimd@gmail.com; 5IRCCS Neuromed, 86077 Pozzilli, Italy; 6Department of Pharmacology, School of Medicine, University of Nevada, Reno, NV 89557, USA; 7Cardiology Unit, Department of Emergency and Critical Care, Tor Vergata Hospital, 00133 Rome, Italy; ruggiero.mango@gmail.com

**Keywords:** arrhythmogenic cardiomyopathy, hypertrophic cardiomyopathy, sudden cardiac death, targeted gene panel, *MYH7* gene

## Abstract

Background: Arrhythmogenic Cardiomyopathy (ACM) is a disease of the cardiac muscle, characterized by frequent ventricular arrhythmias and functional/ structural abnormalities, mainly of the right ventricle. To date, 20 different genes have been associated with ACM and the majority of them encode for desmosomal proteins. In this study, we describe the characterization of two novel variants in *MHY7* gene, segregating in two ACM families. MYH7 encodes for myosin heavy chain β (MHC-β) isoform, involved in cardiac muscle contractility. Method and Results: In family A, the autopsy revealed ACM with biventricular involvement in both the proband and his father. In family B, the proband had been diagnosed as affected by ACM and implanted with implantable cardioverter defibrillator (ICD), due to ECG evidence of monomorphic ventricular tachycardia after syncope. After clinical evaluation, a molecular diagnosis was performed using a NGS custom panel. The two novel variants identified predicted damaging, located in a highly conserved domain: c. 2630T>C is not described while c.2609G>A has a frequency of 0.00000398. In silico analyses evaluated the docking characteristics between proteins using the Haddock2.2 webserver. Conclusions: Our results reveal two variants in sarcomeric genes to be the molecular cause of ACM, further increasing the genetic heterogeneity of the disease; in fact, sarcomeric variants are usually associated with HCM phenotype. Studies on the role of sarcomere genes in the pathogenesis of ACM are surely recommended in those ACM patients negative for desmosomal mutation screening.

## 1. Introduction

Arrhythmogenic Cardiomyopathy (ACM) is a genetic disease of the cardiac muscle, characterized by frequent ventricular arrhythmias and functional and structural abnormalities, mainly of the right ventricle [1,2]. ACM is a cause of Sudden Cardiac Death (SCD) in the young and in athletes [3]. Initially described as Arrhythmogenic Right Ventricular Cardiomyopathy/Dysplasia (ARVC/D), it includes a spectrum of biventricular and left-dominant forms that could be misdiagnosed as dilated cardiomyopathy or myocarditis [4]. The distinctive phenotypic feature is myocardial scar, which underlies global and/or regional ventricular dysfunction and predisposes to potentially lethal scar-related ventricular arrhythmias, regardless of the severity of the systolic ventricular dysfunction [5,6].

The prevalence of the disease is between 1:1000 and 1:5000, depending on the population. The disease shows variable expressivity and reduced, age-related, penetrance. Clinical symptoms typically present in the third to fourth decades of life, with arrhythmic manifestations generally preceding structural features [7].

The disease segregates as autosomal dominant and is genetically heterogeneous; in fact, only 60% of the familial forms can be characterized by genetic tests [8,9,10].

To date, 20 different genes have been associated with ACM [7,10,11,12,13], and the majority of them encode for desmosomal proteins, such as desmoplakin (*DSP*), plakophilin-2 (*PKP2*), desmoglein-2 (*DSG2*), desmocollin-2 (*DSC2*) and plakoglobin (*JUP*) [8,14,15,16,17,18,19]. Cases with a recessive trait of inheritance have also been reported, either associated or not with skin/hair abnormalities [20].

The diagnostic application of Next Generation Sequencing (NGS), now routinely adopted, has surely speeded up the search for genetic variants of such complex and heterogenous phenotypes [21], increasing the possibility of identifying nucleotide variants that are potentially pathogenic. In fact, several recent papers describe novel overlapping complex phenotypes for which NGS analysis has solved their molecular characterization [22,23]. Specifically, in ACM pathogenic variants in sarcomeric genes or in genes responsible for hyperthrophic cardiomyopathy (HCM) have been reported [24,25].

In this study, we describe the characterization of two novel variants in the *MHY7* gene, segregating in two Italian ACM families. *MYH7* gene encodes for myosin heavy chain β (MHC-β) isoform. When mutated, it usually causes structural Cardiomyopathies, such as Hypertrophic (HCM; MIM 192600), Restrictive (RCM), Dilated (DCM; MIM 115200), or Left Ventricular Non-Compaction (LVNC; MIM 613426). Molecular diagnosis has been performed after a detailed clinical evaluation, using a target panel including 70 genes associated with SCD.

## 2. Materials and Methods

### 2.1. Patients

This study included two families (in total eight members) enrolled by the Medical Genetics Unit, Tor Vergata Hospital (Rome, Italy). Informed consent was obtained from each patient after the genetic counselling.

#### 2.1.1. Family A

A 31-year-old female patient (III-2; Figure 1A) presented to clinical evaluation complaining of palpitations. The father had died suddenly at the age of 61 and autopsy examination revealed ACM with biventricular involvement (II-2). The ECG of the proband showed diffuse low-voltages of the QRS and diffuse repolarization abnormalities with negative T waves in lateral leads (Figure 1B). Echocardiographic examination showed a mildly reduced left ventricular ejection fraction (EF = 47%) with septal hypokinesia and apical dyskinesia. The right ventricle had normal size but had a mildly reduced global function with hypokinesia of the inferior-apical segment and aneurysm at the right ventricular outflow tract (RVOT) (Figure 1C). Cardiac Magnetic Resonance confirmed echo findings and reported the presence of multiple foci of late gadolinium enhancement within the left ventricle mostly at the apex, which showed trans-mural involvement. The 24-h ambulatory ECG monitoring showed frequent and complex premature ventricular contraction and non-sustained ventricular tachycardia (NSVT) (Figure 1D, Table 1).

A clinical diagnosis of ACM with biventricular involvement was elaborated and the patient referred for genetic testing. After variant identification, the test was extended to the mother (II-3) and two siblings of the proband (III-3, III-4). Only subject III-3 was heterozygous for the variant. Similar to the proband, subject III-3 had a positive history of recurrent palpitations. Her previous transthoracic echocardiography was reported normal, with a basal ECG reporting low voltage in DI and aVL leads. Although adequately informed about the genetic test results, she denied any cardiological re-evaluation.

The subject III-4 underwent extensively cardiological examination (basal ECG, transthoracic echocardiography, ECG holster monitoring, stress test and cardiac MRI) and no pathological conditions were reported.

#### 2.1.2. Family B

A 59-year-old man (II-3; Figure 2A) with no family history for cardiomyopathies came to our attention to investigate the possible genetic causes underlying his cardiovascular problems (Figure 2A). During 2005, he had a syncopal episode with documented wide QRS complex tachycardia at ECG treated with amiodarone (Figure 2B). Two years later, he was implanted with ICD because of ECG evidence of monomorphic ventricular tachycardia, after syncope (Table 1). The basal ECG recording shows the presence of an epsilon wave in V3–V4, an inverted T wave from V3 to V6, and a Q wave in V5–V6 (Figure 2C). Echocardiographic examination showed dilated right ventricle trabeculation of the free wall, EF: 60%, and mitral valve prolapse with mild regurgitation. Cardiac MRI showed replacement of the ventricular myocardium with fibro-fatty tissue. He had no family history of SCD. Genetic testing in the patient revealed a variant also identified in one of three daughters (III-8), who had some syncopal episodes. She had implanted LRI, while echocardiogram and MRI were in the normal range. She had also been diagnosed with Holt-Horam Syndrome during childhood. The other two (III-6 and III-7) were clinically asymptomatic and negative to the clinical cardiovascular screening.

### 2.2. Next-Generation Sequencing (NGS)

We designed a custom panel for SCD using Ion Ampliseq Designer software (Ampliseq.com; March 2019) (Thermofisher, Waltham, MA, USA), selecting the coding sequence regions for 70 genes [26], for a total of 2002 amplicons.

To generate the libraries, 30 ng of gDNA were used by Ion Ampliseq Library Kit v2.0 (Thermofisher, Waltham, MA, USA), following the manufacturer’s instructions. Libraries were indexed using the Ion Xpress Barcode Adapter Kit. After dilution of all samples at 100 pM, amplicon libraries were pooled for emulsion PCR on Ion Chef according to manufacturer’s instructions. The template was sequenced on the Ion Torrent S5 platform using the 530 chip.

After the run, samples were analyzed using Torrent Suite to obtain Variant Calling Format (VCF) and Binary Alignment Map (BAM) files, and to analyze the quality of the run in order to consider the number of sequencing for each sample and the mean depth of coverage. We analyzed the BAM files with IGV (Integrative Genome Viewer) (www.broadinstitute.org/igv; Accessed on September 2019) [27] to verify the real coverage. VCF files were analyzed in Ion Reporter (wannovar.usc.edu). The first time, only desmosomal genes, associated with ACM (Appendix A), were evaluated by a pipeline created in our lab. The analysis was subsequently extended to all 70 genes included in the panel, after which a further specific informed consent was obtained.

Genetic variants were evaluated based on their frequencies (MAF ≤ 0.01%) and their presence in difference databases, such as ClinVar (https://www.ncbi.nlm.nih.gov/clinvar/, Accessed on January 2020), dbSNP (https://www.ncbi.nlm.nih.gov/snp/), GnomAD (https://gnomad.broadinstitute.org/, Accessed on January 2020), and the Human Gene Mutation Database (http://www.hgmd.cf.ac.uk/, Accessed on January 2020). In silico tools such as Mutation Taster (http://www.mutationtaster.org/, January 2020), Provean (http://provean.jcvi.org/, Accessed on January 2020), VarSome (https://varsome.com/, Accessed on January 2020) and Human Splicing Finder (www.umd.be › HSF, Accessed on January 2020) were used.

Taking into account that allele frequency of ACM in the general population spans from 0.01% and 0.025%, only genetic variants presenting an allele frequency (MAF) equal to or lower than 0.01% in the Genome Aggregation Database (gnomAD) (gnomad.broadinstitute.org Accessed on January 2020) and predicted to be damaging by in silico pathogenicity tools were considered.

### 2.3. In Silico Analysis

Variants on MYH7 were mapped onto the protein structures of the protein alone and complexed as a homodimer and hetero-dimer with MYL2 and MYL3, both X-ray and homology-modeled structures (PDBcode: 2fxo, 2fxm, 5tby and 3dtp), through the DsysMap [28] database (http://dsysmap.irbbarcelona.org/, Accessed on January 2020); both the structure of the protein and of its complexes with other proteins were collected. An additional model of the interaction between the MYH7 homodimer and MYL2/MYL3 proteins was built using as template the PDB structure 6z47, using SWISS-MODEL with a GMQE value of 0.36. FoldX [29] was used to determine the difference in folding free energy (ΔΔG) between the wild type and the mutant protein. First, the PDB structures were repaired with the RepairPDB module of FoldX in order to correct residues with bad torsion angles, or Van der Waals clashes or bad total energy. A 3D model was created with the BuildModel module for both the wild type and mutant proteins. The Stability module was then applied to wild type and mutant proteins obtaining a ΔG for both proteins. The AnalyseComplex was applied on protein–protein complexes in order to obtain the interaction energy for both the wild-type and mutant interactions.

The docking between proteins was performed using the Haddock2.2 webserver (http://milou.science.uu.nl/services/HADDOCK2.2/; Easy Interface service, Accessed on April 2020), which allows information-driven flexible docking. The active residues on the interface were determined previously with FoldX, while passive residues (those surrounding the active residues) were automatically detected. A score that takes into account different energies (Van der Waals, desolvation and electrostatic energies) is provided at the end of the docking analysis. A three-step docking (rigid-body energy minimization, semi-flexible refinement in torsion angle space, and final refinement in explicit solvent refinement) is used to produce a maximum of 200 models of the interaction, which are finally clustered into different solutions. Only the cluster with the best Haddock score is considered.

## 3. Results

### 3.1. Molecular Characterization

Two different *MYH7* variants, one for each ACM patient, were identified in heterozygosity.

These variants were both located within exon 22 on the neck of the protein, in the N-terminal region of the rod, termed subfragment-2 (S2), which joins the heads at the neck region (Figure 3A).

The *MYH7* gene encodes for a myosin heavy chain β (MHC-β) isoform (slow twitch) expressed primarily in the heart, but also in skeletal muscles (type I fibers). This isoform is distinct from the fast isoform of cardiac myosin heavy chain, *MYH6*, referred to as MHC-α. MHC-β is the major protein comprising the thick filament in cardiac muscle and plays a major role in cardiac muscle contraction [30].

In Family A, the missense substitution c.2630T>C (p.Met877Thr) leads to a Methionine to Threonine substitution at amino acid position 877 (Figure 3B). Methionine contains an α-amino group, a carboxyl group and an S-methyl thioether side chain, classifying it as a nonpolar, aliphatic amino acid, while Threonine contains an α-amino group, a carboxyl group and a side chain containing a hydroxyl group, making it a polar, uncharged amino acid. The variant has never been described thus far, and its in silico analysis classifies it as damaging (Mutation taster: disease causing; Provean: deleterious; SIFT: damaging) (Table 2).

The level of conservation at a locus is a strong predictor of the deleteriousness of any change; thus, we compared 10 different eukaryotes organism (*Pan troglodytes, Macaca mulatta, Felis catus, Mus musculus, Gallus gallus, Takifugu rubripes, Danio. rerio, Drosophila melanogaster, Caenorhabditis. elegans, Xenopus tropicalis*) (Table 3) and the methionine results to be conserved in all species were studied. No other variant was found in all genes sequenced with the panel used.

In addition to c.2609G>A, p.Arg870His, that we identified as a novel one in family B (Figure 3C), the same amino acidic residue has been already involved for another variant associated with SCD [24,31]. The variant, leading to an Arginine to Histidine substitution in exon 22, has been previously described as a pathogenic mutation for Hypertrophic Cardiomyopathy (HCM; rs36211715) [32]. Pathogenicity predictors such as Mutation Taster, Provean and SIFT classify it as disease causing, deleterious, and damaging, respectively (Table 2). The variant does not have any gnomAD exome and genome entries. After a comparison of this locus between 10 different eukaryote organisms (Table 4), the Arginine results to be conserved in all species were studied.

### 3.2. In Silico Characterization

To evaluate the function and impact of the Met877Thr variant, an in silico approach was used on different protein structures representing the MYH7 homodimer and the complex interaction between MYH7 and MYL2 and between MYH7 and MYL3. The patient is heterozygous for this missense variant, but no variants within the other two proteins are present.

The stability and binding energy of these protein complexes was evaluated with FoldX (see the Methods section), which provides an empirical force field for the evaluation of the impact of mutations on proteins. The analysis of the impact of the Met877Thr on the stability of MYH7 alone (PDB structure 5tby) did not highlight any significant change (ΔΔG of +0.268 kcal/mol) (Figure 4A). However, the impact of the mutation on the MYH7 homodimer (four PDB structures: 2fxo, 2fxm, 5tby and 3dtp) revealed a highly destabilizing role (ΔΔG of 6.104 ± 0.884 kcal/mol), which is also reflected in the difference in interaction energy between the two monomers (ΔΔG of 4.140 ± 0.383 kcal/mol).

The complexes between MYH7 and the other two proteins are destabilized by the Met877Thr variant in terms of stability (ΔΔG of 1.134 ± 0.643 kcal/mol for the MYH7-MYL3 interaction and ΔΔG of 1.106 ± 0.656 kcal/mol for the MYH7-MYL2 interaction); however, their binding is not significantly affected.

Given the highly destabilizing effect of the Met877Thr variant on the homodimer of MYH7, we investigated its role further by docking the two monomers with Haddock, a software package for information-driven flexible docking. The measured Δ HaddockScore between the wild-type and the mutant was +10.4, meaning that the mutant disrupts the interaction interface, especially through a worse desolvation energy in the mutant state (Appendix A), possibly hinting at a shift of the monomer–dimer equilibrium towards the monomer state and a weaker dimer interaction.

The Met877thr mutation takes place in the middle of three hydrophobic residues (Leu880, Leu881 and Leu873), where the wild-type Methionine does not form H-bonds with other residues, while its Threonine variant forms a novel H-bond with the background of Leu873 and Glu874.

Regarding the impact of the Arg870His variant on MYH7, it resulted in a mild destabilization of the monomer structure (ΔΔG of +0.499 kcal/mol) and in a destabilization of the homodimer both in terms of stability (ΔΔG of +2.042 ± 1.889 kcal/mol) and binding energy between the two monomers (ΔΔG of +2.928 ± 0.771 kcal/mol) (Figure 4B).

Arg870His affects a highly charged region of the protein, with Arg870 interacting with Glu867, Lys871 and Glu875 on the same MYH7 monomer and Arg869 of the other MYH7 monomer. The Histidine variant in this position gains an H-bond with Glu867. Gln176 and Glu177 of MYL3 are the residues in close contact with these mutants. Again, we were able to map this variant onto the interaction interface between MYH7 and MYL3 and between MYH7 and MYL2, with a slightly destabilizing effect on the stability of the structure in both cases (ΔΔG of +0.719 ± 0.272 kcal/mol and of +0.425 ± 0.016 kcal/mol respectively), but no relevant effect on their binding (~ +0.1 kcal/mol). Many of these observations were confirmed when modeling the MYH7 homodimer interacting with MYL2 and MYL3 (as in the 5tby PDB structure) using as template the structure of the shutdown state of myosin-2 from *M.gallopavo* (PDB structure 6z47). Met877Thr had a destabilizing effect for both the stability of the structure (+2.6 kcal/mol) and the interaction between the two MYH7 monomers (+0.8 kcal/mol). The Arg870His mutation was also confirmed to have a destabilizing effect on the interaction between the two MYH7 monomers (+1.85 kcal/mol), but had no noticeable effect on the stability of the overall structure. The effect of both mutants on the interaction energy between MYH7 and MYL2/MYL3 could not be confirmed using this alternative template.

## 4. Discussion

Arrhythmogenic Cardiomyopathy (ACM) is a devastating disease, characterized by significant clinical and genetic heterogeneity. In fact, clinical presentation and disease course, including age of onset, severity of patient symptoms, and adverse outcomes are highly variable and can substantially differ within the same affected family, due to the complex genetic and epigenetic background. The complexity likely reflects the considerable heterogeneity of genes and allelic variants, and the influences of background genotypes, environmental exposures, and lifestyle.

Thus, ACM diagnosis remains challenging, considering that the first symptom of the disease is often sudden cardiac death. For this reason, genetic analysis represents an important support, allowing the pre-symptomatic identification of at-risk subjects and affected relatives. Moreover, *“genotype-positive, phenotype-negative”* individuals can be individuated, lacking hypertrophy or dilatation, but exhibiting cardiac abnormalities that provide insights into the earliest biomechanical defects that link pathogenic genotypes to cardiac dysfunction.

Various genetic mutations have been associated with ACM, but these cannot account for the entire spectrum of disease expression. Therefore, epigenetic and environmental factors may also act as disease modulators. Thus, the identification of novel genes involved in ACM would allow a more precise molecular and clinical diagnosis.

Several genetic loci have been reported to be associated with ACM, usually involved in hypertrophic and dilated cardiomyopathy [33]. Recently, the identification of sarcomeric variants in ACM probands (4.3% of individuals) was also reported [25].

*MYH7* gene encodes for a myosin heavy chain β (MHC-β) isoform expressed primarily in the heart, predominantly in the cardiac ventricle and in type 1 skeletal muscle fibers. This isoform is distinct from the fast isoform of cardiac myosin heavy chain, *MYH6*. MHC-β is the major protein comprising the thick filament in cardiac muscle and plays a major role in cardiac muscle contraction. It is a key component of the sarcomere and exerts its function through extensive protein–protein interactions, with the actin fiber tracks (thin filaments) and other myosin. 

Mutations (more than 200) in the *MYH7* gene are usually associated with several structural Cardiomyopathies, such as Hypertrophic (HCM; MIM 192600), Restrictive (RCM), Dilated (DCM; MIM 115200), or Left Ventricular Non-Compaction (LVNC; MIM 613426), as well as myosin storage myopathy (MIM 608358) [33,34,35]. Moreover, deletions or missense mutations exclusively located within exon 32 to 36 cause Laing distal myopathy (MIM 160500) [35,36,37,38].

In this study, we report the molecular and clinical characterization of two patients affected by ACM and heterozygotes for two different variants in the *MYH7* gene. Both variants lie in the head region of the protein (aa 181–937) and are located within the same functional domain, the S2 domain, potentially affecting neck flexibility during contraction [39].

Both variants are rare and predict damaging. They are rare in the population (c. 2630T>C is not described in any database, while c.2609G>A has a frequency of 0.00000398) and are located in a highly conserved domain.

The analysis of the impact of the Met877Thr on the stability of the MYH7 monomer alone did not highlight any significant change, but the mutated MYH7 homodimer revealed a highly destabilizing role, which is also reflected on the difference in interaction energy between the two monomers. The complexes between MYH7 and the other two proteins, MYL2 and MYL3, are destabilized by the Met877Thr.

The analysis of the other variant, the Arg870His, also resulted in a mild destabilization of the monomer structure, but in a strong destabilization of the homodimer, both in terms of stability and binding energy between the two monomers. The stability of the structure interaction between MYH7 and MYL3 and between MYH7 and MYL2 resulted in be moderately decreased, but no relevant effect on their binding has been revealed. Therefore, in silico analysis, which accounted for static and dynamic energetic evaluation of the complexes involving MYH7, highlighted a deleterious role for both variants.

Given the roles played by MYH7 in cardiac tissues, it is reasonable to hypothesize that the variants identified in our ACM patients could lead to a pathogenic mechanism. In particular, considering the location of the mutations in the coiled-coil rod region, the most likely pathogenic effects would be the impairment of the incorporation of myosin into the myofilaments or the binding to titin [33]. Surely, further in vitro functional studies can support our hypothesis and investigate the functional role of these variants in the function of the sarcomere and the desmosome, and in the pathogenesis of ACM.

Our results report the presence of two variants in sarcomeric genes usually associated with the HCM phenotype and not with ACM. These data further confirm and extend the genetic heterogeneity of ACM and reveal an overlap with phenotypical signs of other cardiomyopathies [21,25].

This evidence suggests that the inclusion of sarcomere genes (*ACTC1, MYBPC3, MYH7, MYL2, MYL3, TNNC1, TNNI3, TNNT2* and *TPM1*) in the classical desmosomal gene screening during molecular characterization of ACM patients should benefit the percentage of patients positive to molecular characterization.

Considering these overlaps, our study aimed to look for new potential gene–phenotype associations with a view to expanding the mutational spectrum underlying cardiomyopathies, and thus provide better diagnosis and clinical management.

## 5. Conclusions

Due to the dramatically increased number of variants identified with NGS approaches and the frequent identification of missense variants in ACM genes in the general population (16%) [40], it is of primary importance that the final interpretation of the genetic results be carried out in close collaboration among experienced professionals at a recognized center. Adequate pre- and post-genetic test counselling for the patients and their relatives is also essential; at the same time, the genetic heterogeneity, incomplete penetrance and variable expressivity related to the inherited cardiomyopathies heighten the complexity of the genetic counselling. This underlines the pressing need for a better understanding of the genotype–phenotype correlation and the modifier factors involved in modulating the clinical phenotype in patients affected with inherited cardiac diseases [24].

## Figures and Tables

**Figure 1 genes-12-00793-f001:**
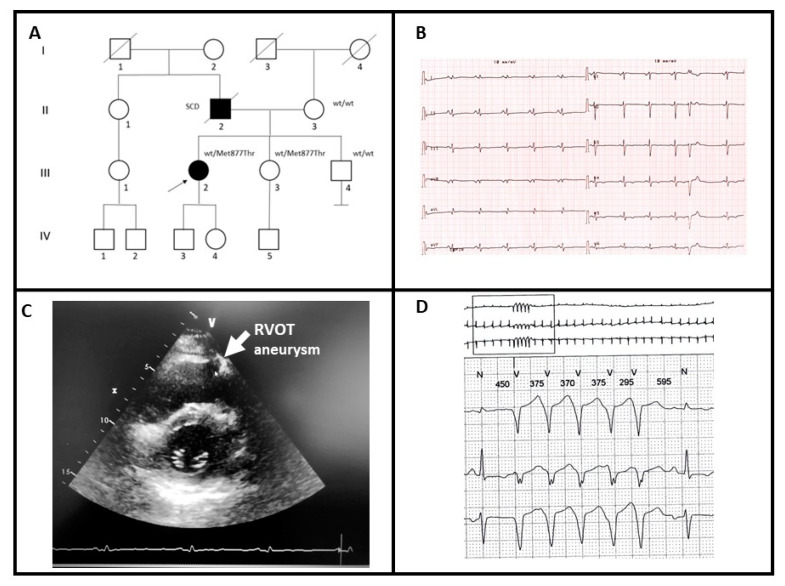
(**A**) Pedigree structure of family A, arrow indicates the proband affected (III-2); Met877Thr is the variant identified within *MYH7* gene; (**B**) basal ECG of the proband; (**C**) transthoracic echocardiography of the proband, the arrow shows aneurysm of right ventricular outflow tract (RVOT); (**D**) 24-h ambulatory ECG monitoring showing NSVT.

**Figure 2 genes-12-00793-f002:**
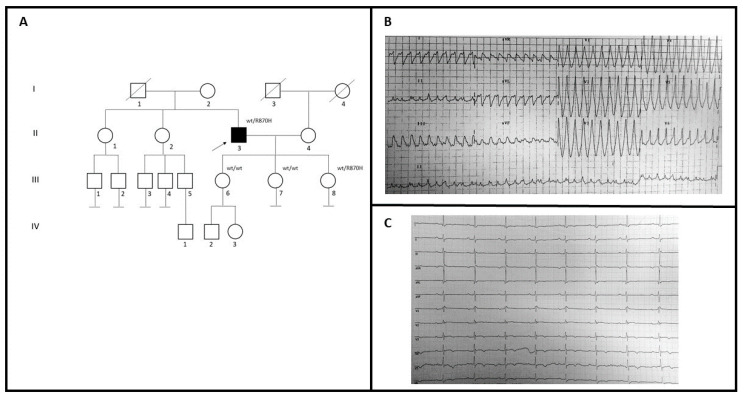
(**A**) Pedigree structure of family B, arrow indicates the proband affected (II-3); R870H is the variant identified within the *MYH7* gene; (**B**) 12-lead ECG showing a wide QRS complex tachycardia; (**C**) basal ECG of the proband.

**Figure 3 genes-12-00793-f003:**
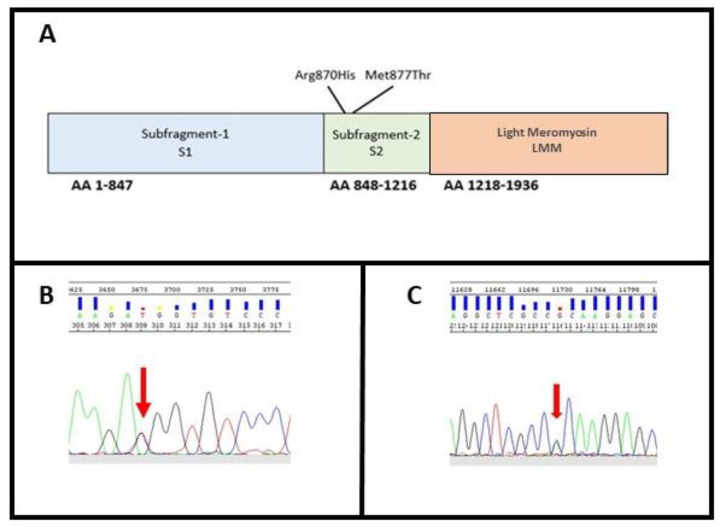
(**A**) MYH7 domain; (**B**) identification of variant M877T in *MYH7*; (**C**) identification of variant R870H in *MYH7*. The arrows indicate the position of the variants identified in heterozygosity.

**Figure 4 genes-12-00793-f004:**
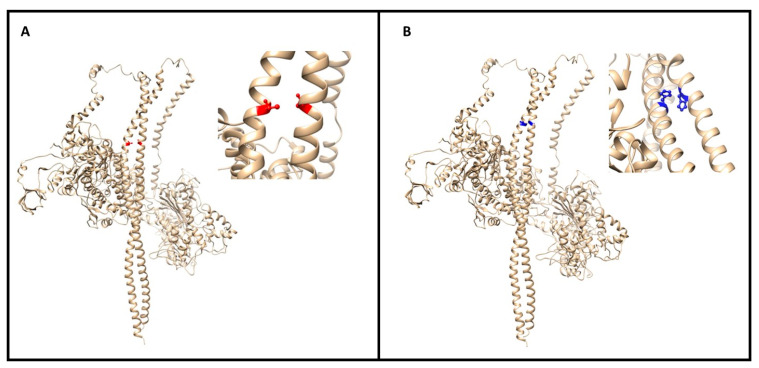
(**A**) The image (made with UCSF Chimera of the myosin dimer (PDB code 3dtp) with the mutation (M877T) colored in red; a close-up of the affected region is reported on the right. The mutations map at the interface between the two long α helices. (**B**) Global and closeup view of the MYH7 dimer (PDB code 3dtp) of the mutation site (R870H in blue). The involved residue positions map on the interaction interface between two α helices of the homodimer.

**Table 1 genes-12-00793-t001:** Clinical information.

Fam	Sex	Age (y)	Family History	Symptoms	ECG	Contrast-Enhanced CMR	Further Examinations	Medical History and FU
1	F	38	Father: scd when 60 years with autoptic diagnosis of AC. Paternal grandfather: SCD when 55 years Two brothers of the paternal grandfather: SCD at 49 and 64 years respectively. Sister: biventricular AC; AICD recipient	Frequent and complex PVB during exercise. NYHA I class.	Diffuse low QRS voltages.Epsilon wave in V1-V3 leads. TWI in 2-V6 leads.	RV dysfunction (RVEF 35%). LV dysfunction (LVEF 41%); apical hypokinesis; infero-distal wall akinesis. Subepicardial mid-basal anterior LV wall LGE; subepi-intramyocardial infero-lateral LV wall LGE; transmural LV apex LGE	Normal coronary computed tomography. ECG Holter (2019): 4693 polymorphic PVBs (prevalent origin from the RVOT9; 78 ventricular couplets	Subcutaneous AICD implanted for primary prevention (2014). No SVT in treatment with sotalol during 6 y of FU.
1	F	37	Father: scd at 60 years with autoptic diagnosis of AC. Paternal grandfather: SCD at 55 years. Two brothers of the paternal grandfather: SCD at 49 and 64 years respectively. Sister: biventricular AC; AICD recipient	Palpitations. NYHA I class.	Low QRS voltages in DI and avL leads	Not done	TTE (2019): normal LVEF (65%); normal RVd2 diameter (23 mm): mild mitral, tricuspidal and pulmonary regurgitation	No syncopal or major arrhythmic episodes during 2 y of FU.
2	M	52	Negative for *SCD/AC*	Syncopal episodes and SVT (2005).NYHA II class.	Diffuse low QRS voltagesEpsilon wave in V1-V3 leads TWI in V2-V6 leads.	RV dysfunction (RVEF 35%). LV dysfunction (LVEF 41%); apical hypokinesis; infero-distal wall akinesis. Subepicardial mid-basal anterior LV wall LGE; subepi-intramyocardial infero-lateral LV wall LGE; transmural apex LGE	TTE (2007): normal LVEF (60%); enlarged RV; mitral valve prolapsed with mild regurgitation. Coronary angiography: normal (2005). Positive EPS (2007).	Dual-chamber AICD implanted (2007). Previous appropriate AICD shock due to SVT (2007). No further SVT/FV during 13 y of FU
2	F	18	Father: biventricular AC, AICD recipient	Holt-Oram syndrome;Type II Mellitus diabetes; Syncopal episodes; Palpitations;NYHA I class	Normal	CMR (2019): normal; no LGE	TTE (2019): normal. Holter ECG (2020): no ventricular arrhythmias	ILR implantations; episodes of sinus tachycardia during palpitations and during syncopal episodes (1 y of FU)

**Table 2 genes-12-00793-t002:** Prediction data for variant c.2630T>C e c.2609G>A in *MYH7*.

Variant	c.2630T>C	c.2609G>A
ACMG classification	Likely pathogenic	Pathogenic
DANN	0.9858	0.9996
MutationTaster	Disease causing	Disease causing
FATHMM	Damaging	Damaging
MetaSVM	Damaging	Damaging
GERP	4.73	4.73
SIFT	Damaging	Damaging
Provean	Damaging	Damaging
Frequency	/	0.00000398

**Table 3 genes-12-00793-t003:** Multiple protein sequence alignment of amino acid 877 in *MYH7*.

Species	Match	AA	Alignment
Human		877	SEARRKELEEK**M**VSLLQEKNDLQL
Human mutated	not conserved	877	SEARRKELEEK**T**VSLLQEKNDLQ
P. troglodytes	no homologue		
M. mulatta	all identical	869	SEARRKELEEK**M**VSLLQEKNDLQ
F. catus	all identical	810	SEARRKELEEK**M**VSLLQEKNDLQ
M. musculus	all identical	877	SEARRKELEEK**M**VSLLQEKNDLQ
G. gallus	all identical	883	SEARRKELEEK**M**VSLLQEKNDLQ
T. rubripes	no homologue		
D. rerio	all identical	879	SEARRKELEEK**M**VSLLQEKNDLQ
D. melanogaster	no homologue		
C. elegans	no homologue		
X. tropicalis	all identical	877	SEARRKELEEK**M**VSLLQEKNDLQ

Bold indicate the ammino acid mutated.

**Table 4 genes-12-00793-t004:** Multiple protein sequence alignment of amino acid 870 in *MYH7*.

Species	Match	AA	Alignment
Human		870	LKEALEKSEAR**R**KELEEKMVSLLQ
Human mutated	not conserved	870	LKEALEKSEAR**H**KELEEKMVSLL
P. troglodytes	no homologue		
M. mulatta	all identical	862	LKEALEKSEAR**R**KELEEKMVSLL
F. catus	all identical	803	LKEALEKSEAR**R**KELEEKMVSLL
M. musculus	all identical	870	LKEALEKSEAR**R**KELEEKMVSLL
G. gallus	all identical	876	LKEALEKSEAR**R**KELEEKMVSLL
T. rubripes	no homologue		
D. rerio	all identical	872	LKEALEKSEAR**R**KELEEKMVSLL
D. melanogaster	no homologue		
C. elegans	no homologue		
X. tropicalis	all identical	870	LKEALEKSEAR**R**KELEEKMVSLL

Bold indicate the ammino acid mutated.

## Data Availability

Variants have been deposited in ClinVar (https://www.ncbi.nlm.nih.gov/clinvar/). Submission ID: SUB9546196. 27/04/2021

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
