# Peer review of "Variants in MHY7 Gene Cause Arrhythmogenic Cardiomyopathy"

_genes, 2021, doi:10.3390/genes12060793_

Round 1
Reviewer 1 Report
This manuscript characterizes two mutations in the MYH7 gene that correlated with diagnoses of ACM in two different families. It is interesting, because mutations that cause ACM normally affect desmosome associated proteins, this report is the first on myosin mutations. Furthermore, the MYH7 is the most common site of mutations that cause familial hypertrophic cardiomyopathies which also can result in sudden cardiac death in young adults and athletes. The manuscript supports the view that these mutations are indeed the cause of ACM by analyzing the predicted impact of the mutations on the structure of the myosin protein using molecular models. A few clarifications in the presentation of the findings would help to strengthen the arguments made.
- Line 38, reads “the major cause” but should instead say, “a cause” as the cited reference states. There are other conditions that can lead to sudden cardiac arrest in young adults and athletes that are similarly or more prevalent than ACM such as familial hypertrophic cardiomyopathies. Emphasizing the similarities in this outcome as well as the genetic causes of these two diseases would help set the stage for later comparisons between them.
- Figures 1 and 2 would benefit from more annotations of the major points in the legend. For instance, the purposes of the arrows are not defined. It is also not clear what the single echocardiogram image in Figure 1c is demonstrating, since multiple frames would be needed to measure ejection fractions.
- It is not immediately clear why in line 147 the 2fxo structure is referenced in addition to the 2fxm structure, since only 2fxm is wildtype while 2fxo contains the E924K hypertrophic cardiomyopathy mutation. Also, the referenced homology model 5tby is based on a 2 nm resolution smooth muscle myosin model, while a newer 0.63 nm resolution 6Z47 model of the smooth muscle myosin shutdown state suggests a little different interaction between the subfragment-2 and the myosin head and light chains. It might be worth clarifying how your docking simulations could be affected by this and possible future variations on the model.
- In lines 155 and 290, the manuscript refers to calculations of interaction energies to support the view that the mutations destabilize the protein structure. These interaction energies should be presented with a clearer description of which residues on the structure were interacting with the mutations for the measurement.
- Lines 196 and 206 read, “amminoacid” which should be “amino acid”.
- Line 238-9 states, “a strong destabilization 238 of the homodimer both in terms of stability (ΔΔG of +2.042±1.889 kcal/mol)” but it does not appear to be significant given the large error estimate. Is there an appropriate statistical measure of significance for this and similar measurements?
- Line 296-7 reads, “resulted to be slightly destabilized” which should be worded differently.
- Line 313 needs clarification where it reads, “could be increment the rate”.
Author Response
"Please see the attachment."

Reviewer 2 Report
This is an interesting and potentially important manuscript, but there are several problems:
- The genetics and correlated cardiac examination for family A are disappointingly incomplete. The paper must include information regarding whether subject III-3 has heart disease.
- The abstract should include the information that the MYH7 gene encodes for a myosin heavy chain beta (MHC-β) isoform
Author Response
"Please see the attachment."
